# The Protection of Quinoa Protein on the Quality of Pork Patties during Freeze–Thaw Cycles: Physicochemical Properties, Sensory Quality and Protein Oxidative

**DOI:** 10.3390/foods13040522

**Published:** 2024-02-08

**Authors:** Zhiming Meng, Ying Liu, Yueyang Xi, Yingying Dong, Chunbo Cai, Yingchun Zhu, Qi Li

**Affiliations:** 1College of Food Science and Engineering, Shanxi Agricultural University, Jinzhong 030801, China; mengzhiming163@163.com (Z.M.); liuying09172021@163.com (Y.L.); 15993819982@163.com (Y.X.); m18536494921@163.com (Y.D.); 15835563385@163.com (Q.L.); 2College of Animal Science, Shanxi Agricultural University, Jinzhong 030801, China; caichunbo@sxau.edu.cn

**Keywords:** quinoa protein, myofibrillar protein, freeze–thaw cycle, protein oxidative, sensory quality

## Abstract

The present study investigated the impact of quinoa protein (QP) on the physicochemical properties, sensory quality, and oxidative stability of myofibrillar protein (MP) in pork patties during five freeze–thaw (F-T) cycles. It was observed that repeated F-T cycles resulted in a deterioration of pork patty quality; however, the incorporation of QP effectively mitigated these changes. Throughout the F-T cycles, the sensory quality of the QP-treated group consistently surpassed that of the control group. After five F-T cycles, the thiobarbituric acid reactive substance (TBARS) content in the control group was measured at 0.423 mg/kg, whereas it significantly decreased to 0.347 mg/kg in the QP-treated group (*p* < 0.05). Furthermore, QP inclusion led to a decrease in pH and an increase in water-holding capacity (WHC) within pork patties. Following five F-T cycles, Ca^2+^-ATPase activity exhibited a significant increase of 11.10% in the QP-treated group compared to controls (*p* < 0.05). Additionally, supplementation with QP resulted in elevated total sulfhydryl content and reduced carbonyl content, Schiff base content, and dityrosine content within myofibrillar proteins (MPs), indicating its inhibitory effect on MP oxidation. In particular, after five F-T cycles, total sulfhydryl content reached 58.66 nmol/mL for the QP-treated group significantly higher than that observed for controls at 43.65 nmol/mL (*p* < 0.05). While carbonyl content increased from 2.37 nmol/mL to 4.63 nmol/mL between the first and fifth F-T cycle for controls; it only rose from 2.15 nmol/mL to 3.47 nmol/mL in the QP-treated group. The endogenous fluorescence levels were significantly higher (*p* < 0.05) in the QP-treated group compared to controls. In conclusion, the addition of QP enhanced the quality of pork patties and effectively inhibited the oxidative denaturation of MP during F-T cycles.

## 1. Introduction

Frozen storage is a commonly used and effective technique for preserving meat and preventing its deterioration [1]. A low temperature (−18 to −35 °C) slows down a range of physiological activities of microorganisms and enzyme activity in meat, thus extending meat storage time [2]. However, because of deficiencies in the frozen food chain or technology, repeated freeze–thaw (F-T) cycles and temperature variations are common and unavoidable in the frozen storage process. F-T cycles in frozen foods can cause several hazards. Firstly, ice crystals will form during the freezing period. If frozen food undergoes F-T cycles, the size of the ice crystals may increase, causing harm to the cellular texture of the food and affecting its texture and quality [3]. Secondly, repeated F-T cycles can create conditions conducive to the growth of microorganisms. As the temperature changes, the surface ice of the frozen food may melt, providing a habitat for the growth of microorganisms. If the food is not consumed or refrozen in a timely manner after thawing, it may lead to the proliferation of microorganisms, thereby compromising the safety of the food. Furthermore, F-T cycles can result in the loss of nutrients in the food [4]. Finally, repeated F-T cycles can also lead to oxidative spoilage of food [5]. During F-T cycles, oxygen may enter the frozen food, causing oxidative spoilage and affecting the taste and quality of the food.

The degradation deterioration of meat production can be prevented by taking new processing technology such as ultrasound-assisted immersion freezing [6] and magnetic field-assisted freezing [7]. Some freezing methods for frozen foods can also avoid repeated F-T cycles such as high-pressure processing at ultra-low temperatures [8]. However, the addition of cryoprotectants to food is the most widely used method to inhibit repeated F-T cycles [9]. The cryoprotectants typically utilized in frozen foods include sugar, sugar alcohols, and proteins [10]. Sugar, such as glucose, sucrose, and trehalose, can be useful for preserving the texture and flavor of food by lowering its freezing point and limiting the formation of ice crystals. Additionally, sugar can increase the osmotic pressure of food, preventing water loss. Sugar alcohols, such as sorbitol and mannitol, can enhance cell membrane stability and prevent ice crystal formation [11]. Sugar alcohol is a new type of sweetener that has good moisture retention and anti-crystallization properties. Proteins, such as serum albumin and plant protein, can protect the cell membrane and protein, and prevent the forming of ice crystals [12]. However, sugar and sugar alcohols are inconsistent with today’s quest for a healthy diet due to their relatively high sweetness and caloric value [13]. Therefore, protein cryoprotectants are the development trend of the food-freezing industry in the future [14]. The utilization of proteins as cryoprotectants is based on two fundamental principles: firstly, these proteins and their derivatives possess antioxidant properties; secondly, they have the ability to bind to existing ice crystals, thereby inhibiting their further growth and reducing mechanical damage caused by large ice crystals in food. Consequently, their incorporation effectively prevents the oxidative cross-linking aggregation denaturation of MP in food [15]. Wang et al. [16] found that chickpea inhibited the denaturation and oxidation of MP during F-T cycles, resulting in increased MP solubility and enhanced Ca^2+^-ATPase activity. In addition, gels prepared from chickpea protein mixed with surimi also exhibited a more stable microstructure. Lin et al. [17] found that gill hydrolysate had antioxidant cryoprotective effects on surimi, comparable to those of commercial cryoprotectants, and could be used as an alternative to sugar cryoprotectants. Plant protein has great development in the food industry because of its wide availability, sustainability, nutrition and health, green environmental protection, and so on [18]. Plant protein as a cryoprotectant is a sustainable development strategy [19].

Quinoa protein (QP) contains a wide selection of essential amino acids that may fulfill the amino acid requirements of adults [20]. QP possesses good functional features, such as emulsification, foaming, gelation, and solubility. Due to the high nutritional value of QP, scientists have proposed it as a substitute for animal protein [21]. The hybrid products of plant and meat proteins have gained acceptance among consumers [22] because they combine the potential nutritional benefits and the original flavor of both ingredients. In a previous study, different amounts of QP were set to investigate the effects of QP on the quality and digestibility of pork patties. It was found that QP could significantly improve the hardness, elasticity, and water retention of the patties, as well as the digestibility of the patties, with 6% QP addition (based on the weight of the meat) showing the most significant improvement. However, whether QP has a good cryoprotective effect on pork patties and their MP or not has not been explored. Therefore, this experiment was conducted to investigate the effects of QP on the sensory quality, physicochemical properties, and MP oxidation of pork patties during repeated F-T cycles by using pork patties made with QP at 6%, compared with those without QP addition.

In this study, the physicochemical properties (pH, color, WHC, thiobarbituric acid-reactive substances, Ca^2+^-ATPase activity), sensory quality, and MP degeneration degree (total sulfhydryl content, carbonyl content, Schiff base content, endogenous fluorescence intensity, dityrosine content) of pork patties were investigated with the addition of QP during different F-T cycles. Additionally, the degradation and aggregation of MP due to QP addition were monitored by sodium dodecyl sulfate-polyacrylamide gel electrophoresis (SDS-PAGE). The aim of this study was to determine whether QP could improve the sensory properties, physical, and chemical properties of pork patties and inhibit MP denaturation during repeated F-T cycles.

## 2. Materials and Methods

### 2.1. Materials

The fresh pork longissimus dorsi muscle of Jinfen White Pig (6-month-old boar) was bought from the local meat market (Jinzhong, China). The QP (purity ≥ 80%) was purchased from Shaanxi Jiucifang Biotechnology Co., Ltd. (Xi’an, China). Sugar, white pepper, five-spice powder, sesame oil, and corn starch were purchased from Taigu Jiajiali Supermarket (Jinzhong, China).

Analytical grade chemicals, including ether, KCl, MgCl_2_, NaCl, HCl, glycine, and ethylenediaminetetraacetic acid (EDTA), were purchased from Chengdu Kolon Chemical Co., Ltd. (Chengdu, China). Trichloroacetic acid (TCA), urea, 2,4-dinitrophenylhydrazine (DNPH), Kormas Brilliant Blue, thiobarbituric acid (TBA), 1,4-piperazine diethanesulfonic acid (PIPES), 5,5′-Dithiobis-(2-nitrobenzoic acid) (DTNB), sodium pyrophosphate, and Na_2_HPO_4_ were purchased from Tianjin Kaitong Reagent Co., Ltd. (Tianjin, China).

### 2.2. Preparing of Pork Patties

*Pork longissimus dorsi* and fat in a 7:3 ratio, minced in a meat chopper (ZB-5, Huaying Food Machinery Co., Ltd., Langfang, China), and 2.600% NaCl, 0.400% sodium phosphate complex, 0.10% sodium ascorbate, 1.00% sugar, 0.20% white pepper, 0.080% five-spice powder, and 1.20% sesame oil (all materials based on meat weight) were stirred together and 20% ice water was added, then divided into two groups: one acts as the control group (without QP) and the other as the QP-treated group with 6% QP (based on the weight of meat). Pork patties were produced using a mold with a diameter of 7 cm and a thickness of 2 cm. The patties were frozen at −18 °C for 7 days and subsequently thawed at 4 °C until the center temperature achieved 4 °C, following an F-T cycle. This procedure was reiterated 5 times.

### 2.3. Extraction of MP

The extraction of MP was referenced to Li et al. [23] method. Ground meat was mixed with four volumetric portions of precooled extraction buffer (10 mmol/L Na_2_HPO_4_/NaH_2_PO_4_, 0.1 mmol/L NaCl, 2 mmol/L MgCl_2_, 1 mmol/L EGTA, pH = 7.0, 4 °C) using an Ultra-Turrax homogenizer (T-10, IKA, Staufen, Germany) to obtain homogeneous samples (8000× *g*, 30 s). The pellet is then filtered (20 mesh, 0.9 mm) and centrifuged at 4000× *g* at 4 °C for 15 min. There were three repetitions of this procedure. After precipitation, the same sample was homogenized in four times the volume of salt solution (0.1 mol/L NaCl), centrifuged at 4000× *g* for 15 min, and washed three times. The resulting sediment was collected as MP, which was stored at 4 °C for 48 h. The content of pure MP was determined using the bolometer method.

### 2.4. pH

To determine the pH of pork patties, 5 g pork patties were placed in 15 mL of distilled water and homogenized at 12,000 rpm for 1 min. This study was conducted at 25 °C. An acidometer (FE28, Metter Toledo, Shanghai, China) was used to measure the pH of all samples.

### 2.5. Ca^2+^-ATPase Activity

Ca^2+^-ATPase activity was measured by commercially available Ca^2+^-ATPase activity assay kit instructions (Solarbio Biotechnology, Beijing, China). Ca^2+^-ATPase activity was expressed as μmol (Pi)/mg (pro)/h. This study was conducted at 25 °C.

### 2.6. Color

To calculate L*-values, a*-values, and b*-values, a colorimeter (CM-5, Konica Minolta Office System Co., Ltd., Tokyo, Japan) was utilized. This study was conducted at 25 °C. The whiteness (W*) of pork patties was defined using Equation (1) as follows:(1)$$W^{*}=100-\sqrt{{\left(100-L^{*}\right)}^{2}+{a^{*}}^{2}+{b^{*}}^{2}}$$

### 2.7. WHC

#### 2.7.1. Thawing Loss

After thawing the frozen specimens (m_1_) were thawed at 4 °C until the center temperature was 0 °C, and surface moisture was absorbed by filter paper and weighed again (m_2_). This study was conducted at 25 °C. The following Equation (2) was used to calculate the thawing loss of pork patties:(2)$$Thawing\;loss\left(\%\right)=\frac{m_{1}-m_{2}}{m_{1}}\times 100$$

#### 2.7.2. Centrifugal Loss

A 10 g sample (m_3_) was weighed and centrifuged by a high-speed freezing centrifuge (5801R, Eppendorf Co., Ltd., Hamburg, Germany) at 4000× *g* for 15 min at 4 °C and weighed again (m_4_). This study was conducted at 25 °C. The following Equation (3) was used to calculate the centrifugal loss of pork patties:(3)$$Centrifugal\;loss\left(\%\right)=\frac{m_{3}-m_{4}}{m_{3}}\times 100$$

### 2.8. Sensory Evaluation

Ten postgraduate students (5 males and 5 females) specializing in food science conducted the sensory evaluation in the sensory analysis laboratory. The participants received specific training to visually and numerically discern sensory differences in samples, as well as accurately interpret the terminology employed in sensory assessment. Heat the pork patties in an oven at a temperature of 200 °C for 20 min, closely monitoring them during the heating process to ensure they are not overcooked or burnt. After completing the heating process, allow the pork patties to cool slightly before evenly cutting them into pieces measuring 2 × 2 cm. The sensory evaluation was carried out using a 9-point scale, where a score of 9 indicated extreme liking, a score of 1 represented extreme disliking, and a score of 5 denoted neutrality towards the sample. The participants were fully aware that their participation in this experiment was voluntary and complied with all aspects of the “1975 Declaration of Helsinki”.

### 2.9. Thiobarbituric Acid-Reactive Substances (TBARS)

TBARS was determined using the method described by Li et al. [24]. Ground pork patties (5 g) were mixed with 15 mL 7.5% TCA solution, and were homogenized for 30 s. The solution was filtered, the filtrate was collected, and 2.5 mL of filtrate and 2.5 mL of 0.02M TBA solution were added. The mixed solution was heated in water at 100 °C for 40 min followed by rapid cooling to 20 °C. The suspension was vortexed for 1 min with chloroform. Supernatants were centrifuged at 2000 rpm for 10 min and absorbance was measured at 532 nm using a spectrophotometer (RF-5301PC, Shimadzu Corporation, Kyoto, Japan) This study was performed at 25 °C. The Equation (4) was used for the expression of TBARS in pork patties:(4)$$TBARS\left(mg/kg\right)=\frac{A_{532}}{W_{s}}\times 9.48$$
where A_532_ indicates the absorbance value of the solution at 532 nm; W_s_ denotes the weight of pork patties (g); and “9.48” is the derivatization constant obtained from the dilution factor and molar extinction coefficient of the TBA reaction product (152,000 M^−1^·cm^−1^).

### 2.10. Total Sulfhydryl Content

The method of Cai et al. [25] was used to determine the total sulfhydryl content. The absorbance of the MP solution containing DTNB was detected at 412 nm. The total sulfhydryl content was calculated using a molecular absorption constant of 13,600 mol/(L·cm). It was expressed as nmol/mg of protein. This study was conducted at 25 °C.

### 2.11. Carbonyl Content

Carbonyls were determined according to Levine et al. [26]. Carbonyl content of DNPH and MP mixed solutions was determined at 370 nm using spectrophotometer (RF-5301PC, Shimadzu Corporation, Japan) and expressed as nmol DNPH/mg protein. This study was performed at 25 °C.

### 2.12. Schiff Base Content

The method developed by Utrera and Estévez [27] was used to determine the Schiff base content. To measure the fluorescence intensity of the samples, 3 mL of 15 nmol/mL PIPES was added to 1 mL of 1 mg/mL MP solution. The fluorescence photometer (RF-5301PC, Shimadzu Corporation, Japan) was used with an excitation wavelength of 350 nm and an emission wavelength scanning range of 400–500 nm. The study was conducted at 25 °C.

### 2.13. Dityrosine Content

Dityrosine content was detected with reference to the method of Ma et al. [28] with a few modifications. Using a fluorescence photometer (RF-5301PC, Shimadzu Corporation, Japan) with an emission wavelength of 420 nm and an excitation wavelength of 320 nm, the dityrosine content was expressed as the fluorescence intensity of 1 mg/mL MP. This study was conducted at 25 °C.

### 2.14. Endogenous Fluorescence Intensity

Fluorescence intensity of MP solution was measured by Zhang et al. method to ensure objectivity and accuracy [29]. The endogenous fluorescence intensity of MP solution (0.05 mg/mL) was measured with a fluorescence photometer (RF-5301PC, Shimadzu Corporation, Japan) with an excitation wavelength of 295 nm and an excitation scanning range of 310–400 nm. This study was conducted at 25 °C.

### 2.15. SDS-PAGE

SDS-PAGE was performed according to the method described by Li et al. [30] with a few modifications to evaluate protein changes in the samples during repeated freeze–thaw cycles. Briefly, the protein solution was extracted and diluted to a concentration of 5 mg/mL, followed by the addition of the up-sampling buffer. The protein solution was heated in water at 100 °C for 3 min to denature it completely. Then, 20 μL of denatured protein solution was added to the electrophoresis apparatus (DYCZ-24DN, Beijing Liuyi Factory, Beijing, China). Upon completion of the electrophoresis, the resulting gel was stained with 1 mg/mL Coomassie Brilliant Blue R-250 solution for 20 min, and then the gel was decolorized to clear with a mixture of 5% methanol and 7% acetic acid. Finally, the gel was imaged with a gel imager (WD-9413A, Beijing Liuyi Factory, Beijing, China). This study was conducted at 25 °C.

### 2.16. Statistical Analysis

A triplicate analysis of all samples was conducted, with the results presented as mean ± standard deviation (SD). Statistix 8.1 software (SAS Institute Inc., Cary, NC, USA) was employed to statistically analyze the data obtained from the experiments, utilizing the two-way analysis of variance. Duncan’s test was employed to contrast treatment means, with statistically relevant disparities being established at a p-value of less than 0.05. These figures were all plotted using Origin Pro 2018 (Origin Lab Corp., Northampton, MA, USA) and TBtools (South China Agricultural University, Guangzhou, China).

## 3. Results and Discussion

### 3.1. Sensory Evaluation

The results of sensory evaluation for both the control group and the QP-treated group during different F-T cycles were presented in Figure 1A,B. For sensory scores, we used a single nine-point scale sensory evaluation [31]. Generally speaking, a sensory score of more than five means that the food is already acceptable to consumers. During five F-T cycles, the sensory evaluation of most indicators of the control group decreased, indicating that the number of F-T cycles harmed the taste, flavor, and tenderness of the patties (Figure 1A). The total sensory score of the control group in Figure 1B also decreased with the increase in F-T cycles, which further indicated that the F-T cycle reduced the sensory quality of the pork patties. The decrease in texture and tenderness of the pork patties might be due to the loss of water caused by repeated F-T. Additionally, the bad flavor of the patties might be attributed to the excessive oxidation and decomposition of fat, leading to the production of by-products of oxidation including aldehydes, ketones, and alcohols [32]. The sensory quality of pork patties after adding QP was improved (Figure 1A), and the total sensory score decreased slightly after the fifth F-T cycle (Figure 1B). This might be due to the addition of QP to pork patties enhancing the adsorption of other flavor components in meat, thereby improving the flavor quality of pork patties. In addition, the QP-added was cross-linked with the MP in pork patties, which enhanced the water retention and improved the taste and tenderness of pork patties.

### 3.2. pH

One crucial factor to consider while assessing the quality of pork patties is pH. The results of pH changes during several F-T cycles are indicated in Figure 2A. The pH of the control group showed an increasing trend throughout the F-T cycles, but that of the QP-treated group increased during four F-T cycles and then decreased after the fifth F-T cycle. Ebadi et al. [33] also found that the pH of gentian fillets increased with the lengthening of freezing during storage. The rise of pH during storage was caused by the production of proteases to decompose nitrogenous compounds such as proteins and amino acids in meat and the oxidation of proteins and fats during F-T cycles [34]. The pH of the QP-treated group decreased in the fifth F-T cycle, which can be attributed to the continuous degradation of proteins in the meat caused by repeated F-T cycles, resulting in a continuous production of various amino acids. Consequently, there was a higher content of acidic amino acids compared to basic amino acids, leading to a lower pH. The pH of the QP-treated groups was always less than that of the control groups throughout the F-T cycles. QP contains a high amount of acidic amino acids, such as glutamic acid and aspartic acid, which can react during the frozen process to produce a large number of acidic metabolites, thus lowering the pH of pork patties [21].

### 3.3. Ca^2+^-ATPase Activity

Ca^2+^-ATPase activity is one of the indicators of changes in protein properties of meat products during frozen storage, which characterizes the integrity of myosin [35]. As shown in Figure 2B, during five F-T cycles, Ca^2+^-ATPase activity of all control groups and QP-treated groups decreased gradually, decreasing to 20.3% and 18.5%, respectively, compared to the starting value. The drop in Ca^2+^-ATPase activity might have been caused by an increase in ionic strength, which led to the production of disulfide bonds by sulfhydryl oxidation. This resulted in protein aggregation and the growth of ice crystals in the pork patties. This results in protein aggregation and the growth of ice crystals in pork patties, which, in turn, destroys the microstructure of the myosin head region and causes a decrease or even disappearance of the enzyme’s functional region [36]. Jiang and Wu [37] also reported that the Ca^2+^-ATPase activity of MP gradually decreased with increasing frozen time.

The control group had lower Ca^2+^-ATPase activity than the QP-treated group at the same F-T cycle, and the Ca^2+^-ATPase activity of the QP-treated group increased by 11.10% compared with that of the control group during five F-T cycles. The results showed that QP could enhance Ca^2+^-ATPase activity, indicating that QP had cryoprotective effects on pork patties’ myosin during multiple F-T cycle periods. Cen et al. [38] found adding QP Pickering emulsion hindered the formation of large ice crystals from MP gels. The incorporation of QP reduced the time required for pork patties to pass through the region where maximal ice crystal growth occurs [39]. Therefore, QP greatly reduced the irreversible mechanical damage caused by the formation of large ice crystals in the pork patties, thus delaying the degree of oxidation of myosin.

### 3.4. Color

The color of the meat could visually indicate the degree of spoilage [40]. Table 1 displays the color changes in pork patties that underwent various F-T cycles. With increasing F-T cycle number, both the control and QP-treated groups exhibited a significant decrease in a*, while L* and W* were significantly improved (*p* < 0.05). The b*-values increased initially and then decreased. During F-T cycle processing, prolonged oxidation leads to protein denaturation, which activates endogenous enzymes to hydrolyze muscle structures, thus causing muscle fiber fragmentation, resulting in higher exudation of drip from the longest dorsal muscle. And the increase in surface moisture content reinforces light reflectance, causing increased L*-values and W*-values. The decrease in a*-values might be owed to myoglobin denaturation during F-T cycles and irreversible oxidizing to dark-brown high-iron myoglobin. The a*-values of the QP-treated groups were consistently higher than those of the control groups throughout the F-T cycle. This might be because QP prevented the development of large ice crystals during the F-T cycles, maintained cellular identity, and diminished protein oxidation [41]. In the F-T cycle, the W*-values of the QP-treated group were consistently lower than that of the control group, which was mainly due to the fact that the QP itself was light yellow in color, and the adulteration of the QP resulted in the darkening of the color of the pork patties.

### 3.5. WHC

The WHC of pork patties was assessed by centrifugal loss (Figure 2C) and thawing loss (Figure 2D). Both centrifugal loss and thawing loss of pork patties in both control groups and QP-treated groups increased with the extension of the F-T cycle number. Specifically, from the first F-T cycle to the fifth F-T cycle, the centrifugal loss of the control group increased from 12.04% to 28.95%, while the QP-treated group experienced an increase from 6.99% to 23.01%. The thawing loss of the control group and the QP-treated group increased from 0.64% and 0.32% to 1.28% and 1.01%, respectively. The increase in thawing loss and centrifugation loss of pork patties indicates a decrease in the WHC. The changes were in accordance with that of Wang et al. [42], in which the WHC of quick-frozen patties decreased continuously during multiple F-T cycles. Repeated freezing and thawing of ice crystals can disrupt the structural integrity of muscle tissue, leading to a decline in WHC. Furthermore, the control groups showed significantly higher centrifugal and thawing losses compared to the QP-treated groups for all F-T cycles (*p* < 0.05). The changes indicated that incorporating QP significantly increased the WHC of pork patties. Cen et al. [43] also reported that QP Pickering emulsion could improve the WHC of MP gels. The addition of QP to meat patties might improve water retention due to its ability to inhibit or reduce the creation of irregular ice crystals during repeated F-T cycles. This could also reduce the loss of muscle fibers and prevent water loss [44]. Secondly, QP filled the gap between the meat pie gels, and QP and MP cross-linked to form a dense network structure, which enhanced the water retention ability of MP. Furthermore, QP is high in amino acids, particularly hydrophilic ones like glutamic acid and aspartic acid. These amino acids can create hydrogen bonds with water molecules, increasing the protein’s ability to bind to water and improve water retention [21].

### 3.6. TBARS

TBARS was used to evaluate lipid oxidation in frozen meat patties during F-T cycles [45]. Figure 3 shows the changes in TBARS of pork patties subjected to different F-T cycles. Over five F-T cycles, TBARS increased from 0.195 to 0.423 mg/kg and 0.189 to 0.347 mg/kg in both control and QP-treated groups. Utrera et al. [46] also found that frozen storage increased the TABRS of pork patties. During repeated F-T cycles, ice crystals continued to grow, damaging the integrity of muscle cells and causing the release of prooxidants including lipase, which accelerated lipid oxidation. In the first F-T cycle, there was no significant difference in TBARS between the control group and the QP-treated group (*p* < 0.05). However, in the following four F-T cycles, the TBARS of control groups were significantly higher than those of QP-treated groups (*p* < 0.05). Cen et al. [43] found that the addition of QP Pickering emulsion reduced the number of large ice crystals in MP gels, decreased damage to the gel structure, and inhibited the aggravation of the degree of lipid oxidation. In addition, QP contains a range of antioxidant components, including phenolic compounds and vitamin E [47]. These antioxidants can inhibit fat oxidation and decrease the production of free radicals, ultimately reducing the TBARS value.

### 3.7. Total Sulfhydryl Content

The susceptibility of sulfhydryl groups to oxidation to covalent disulfide bonds (-S-S-) is the primary cause of crosslinking between MPs [48]. The inhibitory effect of QP on total sulfhydryl deterioration of the MP of pork patties during F-T cycles is shown in Figure 4A. During five F-T cycles, the total sulfhydryl content of the control group and the QP-treated group gradually decreased to 43.65 and 58.66 nmol/mg, respectively, compared to the starting value. During the frozen storage period, the total sulfhydryl content of shrimp gradually decreased, as discovered by Shi et al. [49]. During the first F-T cycle period, the total sulfhydryl content of the control group was higher than that of the QP-treated group, but the control group had lower total sulfhydryl content than the QP-treated group at the same F-T cycle. The reduction in the total sulfhydryl content observed in the QP-treated group after the first F-T cycle might be attributed to the cross-linking of QP with certain MP, thereby the disulfide bonds were formed. This process resulted in a lower total sulfhydryl content in the original MPs than in the control group. During five F-T cycle processes, the total sulfhydryl content of the QP-treated group was 58.66 nmol/mL, which was higher than that of the control group (43.65 nmol/mL). This result suggested that QP had a significant effect on inhibiting the oxidation of sulfhydryl groups. It was possible that QP interacts with other components in pork patties, forming a complex network structure. This interaction might enhance the stability and frost resistance of the protein, while reducing protein degradation and loss of sulfhydryl groups. Moreover, many studies have shown that QP contains high levels of cysteine [50]. Cysteine is essential for maintaining the activity of thiol enzymes [51]. When the content of cysteine increased, the increase in thiol enzyme activity promoted the formation of sulfhydryl group.

### 3.8. Carbonyl Content

Carbonylation is the most characteristic type of protein oxidation. Figure 4B shows the changes in the carbonyl content of pork patties during repeated F-T. The carbonyl content of all pork patties increases significantly with increasing F-T cycle number (*p* < 0.05). In the control group and the QP-treated group, the carbonyl content levels during five F-T cycles increased from 2.37 and 2.15 to 4.63 and 3.47 nmol/mg, respectively. The increase in carbonyl content was owed to the gradual transformation of water into ice during the freezing process, which decreased the water content and increased the concentration of oxidants, contributing to the deepening of protein oxidation. Utrera et al. [46] concluded the recrystallization phenomenon during the F-T promoted the production of giant ice crystals, which disrupted cellular integrity and accounted for the release of prooxidative iron from cells such as heme iron, trimethylamine oxidative demethylase, and other pro-oxidative substances, then further exacerbated protein oxidation. The QP-treated group had a significant reduction in carbonyl content during various F-T cycles in comparison to the control group (*p* < 0.05). Furthermore, the rate of carbonyl content growth in the QP-treated group decelerated after the third F-T cycle. This suggests that the inclusion of QP considerably postponed the oxidation of pork patties. Firstly, it was possible that the addition of QP prevented the enlargement of ice crystals, limited the damage to the MP structure, and decreased the generation of free radicals. Subsequently, QP might have good stability [21], maintaining its structure and function during freezing and thawing, which reduces protein degradation and carbonyl formation. Cen et al. [43] also reported that the insertion of QP Pickering emulsion caused a significant reduction (*p* < 0.05) in the carbon group content of MP gels.

### 3.9. Schiff Base Content

Schiff base formation occurs when free amino groups react with oxidized reducing sugar and lipids [52]. Schiff base content is commonly used to assess protein oxidation. Figure 4C shows the changes in Schiff base content in pork patties during F-T cycles. In the control group and the QP-treated group, Schiff base content increased with the F-T cycle number, and increased to 2.039 and 1.540, respectively, compared to the starting values, which was consistent with the change in TBARS. It was possible that sequential lipid oxidation and protein degradation furnished carbonyl and free amino groups for Schiff base formation [53]. And Schiff base content of the QP-treated group for different F-T cycles significantly decreased in comparison with the control group (*p* < 0.05). This alteration indicates that the addition of QP had a marked inhibitory effect on MP oxidation during F-T cycling, thereby delaying lipid oxidation of patties (*p* < 0.05).

### 3.10. Dityrosine Content

Under oxidative stress, tyrosine in meat can be converted into dityrosine [28]. The severity of protein oxidation can be evaluated by the dityrosine content, and its higher levels indicate more severe oxidation. Figure 4D shows the changes in the dityrosine content in pork patties during F-T cycles. The dityrosine content, carbonyl content, and Schiff base content of pork patties protein showed a similar trend during the F-T cycle period. Figure 4D shows that the dityrosine content of pork patties increased during the five F-T cycles and changed significantly between the third and fifth cycles (*p* < 0.05). Adding QP reduced the dityrosine content compared to the control, indicating that QP had a good antioxidant effect on MP during F-T cycles.

### 3.11. Endogenous Fluorescence Intensity

The decrease in endogenous fluorescence intensity caused by tryptophan oxidation is usually used as one of the indicators to evaluate the oxidative deterioration of meat protein [54]. As depicted in Figure 4E,F, the endogenous fluorescence intensity gradually decreased with an increase in repeated F-T cycles. From the first to the fifth F-T cycle, there was a significant decrease in the maximum fluorescence intensity (FImax) of the control group and the QP-treated group, decreasing from 306.249 and 340.744 to 140.351 and 297.575, respectively (*p* < 0.05). Additionally, the repeated F-T cycle processes might result in the denaturation of protein in pork patties, causing alterations in its natural structure and arrangement. This process also led to the partial unfolding of the protein, exposing the originally internal tryptophan on the protein’s surface. Consequently, the tryptophan was transferred to a polar surrounding, which caused the endogenous fluorescence intensity to decrease. When QP was added to pork patties, the FImax was higher than those without QP at the same F-T cycle. During the second to fifth F-T cycles of the QP-treated group, the endogenous fluorescence intensity of MP did not change significantly (Figure 4F). Thereby indicating that the addition of QP could inhibit the oxidative denaturation of MP caused by F-T cycles. The incorporation of QP promoted interactions between protein molecules and accelerated the rearrangement and aggregation of MP polar groups. A hydrophobic environment was re-formed inside the aggregates, allowing more fluorescent chromophores to enter into it, resulting in a decrease in the number of fluorescent chromophores exposed to the solvent [37].

### 3.12. SDS-PAGE

Figure 5 shows the variation in SDS-PAGE bands of pork patties MP during different F-T cycles. The distribution of myosin heavy chain and actin remained constant throughout the F-T processes. The bands of the control group weakened as the F-T cycles increased, which indicates that the process of repeated F-T cycles deepened the oxidative degradation of MP in the control group. Multiple F-T cycles might cause some of the liquid water in the meat to turn into solid ice crystals, increasing the solute concentration and accelerating the denaturing degradation of proteins. Tan et al. [55] also reported that a decrease in the level of the band was observed after F-T treatment. The bands of QP-treated samples remained almost unchanged during five F-T cycles. Moreover, after the third F-T cycle, it was clearly observed that the actin band of the control group was narrower than that of the QP-treated group during the same period. This observation further supported the notion that QP treatment effectively inhibited the degradation of MP. The variations were consistent with the results in total sulfhydryl, carbonyl, endogenous fluorescence intensity, Schiff base, and dityrosine levels, which further indicated that QP possessed good antioxidant activity.

### 3.13. Analysis of Relationship

In order to investigate the relationship between the quality characteristics of pork patties and the degree of MP oxidation, a correlation analysis was conducted between the physicochemical characteristics of pork patties and the MP oxidation indexes. The results are shown in Figure 6. It is seen in Figure 6 that the storage quality indexes such as pH, a*-value, centrifugal loss, thawing loss, and Ca^2+^-ATPase had significant (*p* < 0.05) or extremely significant (*p* < 0.01) effects on a series of protein oxidation indexes such as carbonyl, total sulfhydryl, dityrosine, Schiff base, and endogenous fluorescence intensity. Because the level of protein oxidation in muscle was closely related to the meat quality, excessive oxidation of muscle protein leads to the deterioration of meat quality, such as water retention, tenderness, and meat color, resulting in economic losses [5]. pH, centrifugal loss, and thawing loss were positively correlated with Schiff base, carbonyl, and dityrosine, and negatively correlated with total sulfhydryl content and endogenous fluorescence intensity. The changes confirmed that the oxidation level of pork patties affected its storage quality. TBARS had an extremely significant positive correlation with centrifugal loss, thawing loss, carbonyl, dityrosine, and Schiff (*p* < 0.01), and had an extremely significant negative correlation with Ca^2+^-ATPase and total sulfhydryl (*p* < 0.01), and had a significant negative correlation with endogenous fluorescence intensity (*p* < 0.05). These phenomena suggest that the level of lipid oxidation in pork patties was positively correlated with the level of protein oxidation, but negatively correlated with storage quality. Specifically, lipid oxidation intermediates such as hydroxyl radicals and oxidation secondary products such as α- and *β*-polyunsaturated fatty aldehydes could promote protein oxidation. During processing and storage, myoglobin and hemoglobin released a large amount of non-heme iron and ferrous heme, which could further promote lipid oxidation.

## 4. Conclusions

Repeating the F-T cycle caused a sharp decrease in the quality of pork patties and also exacerbated the oxidative denaturation of MP. However, QP incorporation significantly decreased the pH, TBARS, carbonyl content, dityrosine content, and Schiff base content of pork patties (*p* < 0.05), while increasing the a*-value, WHC, Ca^2+^-ATPase activity, total sulfhydryl content, and endogenous fluorescence intensity (*p* < 0.05), and the organoleptic qualities of the QP-treated group were higher than those of the control group. SDS-PAGE analyses also found that the level of bands in the QP-treated group was higher than that in the control group. These changes indicate that the addition of QP improved the quality of pork patties and delayed the oxidative deterioration of MP. This is mostly due to the fact that the addition of QP can inhibit the formation of large ice crystals during the F-T cycle or increase the antioxidant and other biological activities of QP itself. In summary, the incorporation of QP inhibited the degradation of pork patty quality by repeated F-T cycles, delayed the oxidative denaturation of pork patty MP, and QP could be utilized as a food antifreeze protectant. Meanwhile, QP was an excellent and sustainable alternative source of animal protein, and its addition could improve the product characteristics of pork patties, which is important for the future development and protection of the food industry.

## Figures and Tables

**Figure 1 foods-13-00522-f001:**
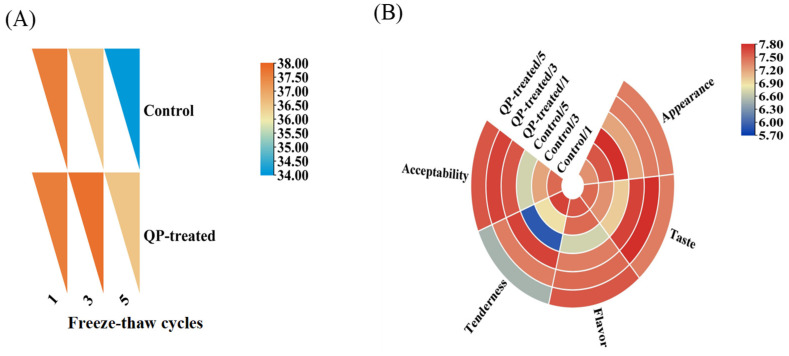
Effect of QP addition on sensory evaluation (**A**) and total sensory score (**B**) of low-sodium pork patties during different F-T cycles.

**Figure 2 foods-13-00522-f002:**
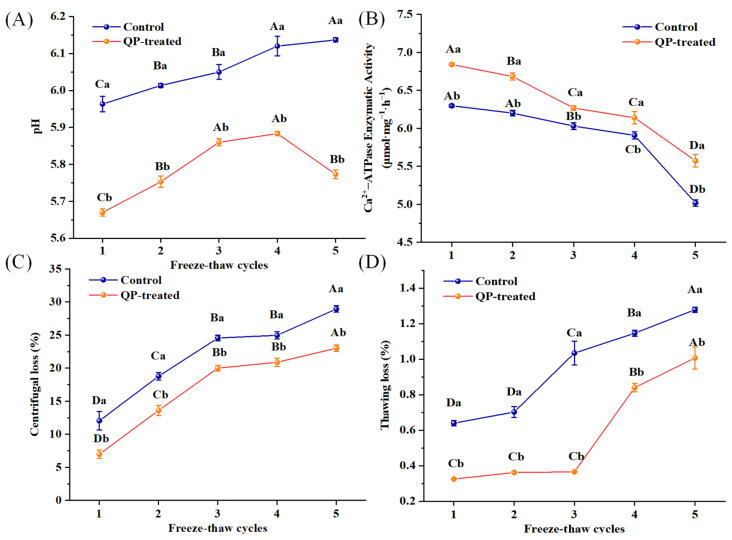
Effect of QP addition on pH (**A**), Ca^2+^-ATPase activity (**B**), centrifugal loss (**C**), and thawing loss (**D**) of low-sodium pork patties during different F-T cycles. Note: QP refers to quinoa protein. Different lowercase letters indicate a significant difference between the treatment groups within the same F-T cycle (*p* < 0.05); different uppercase letters indicate a significant difference between the F-T cycles within the same treatment group (*p* < 0.05).

**Figure 3 foods-13-00522-f003:**
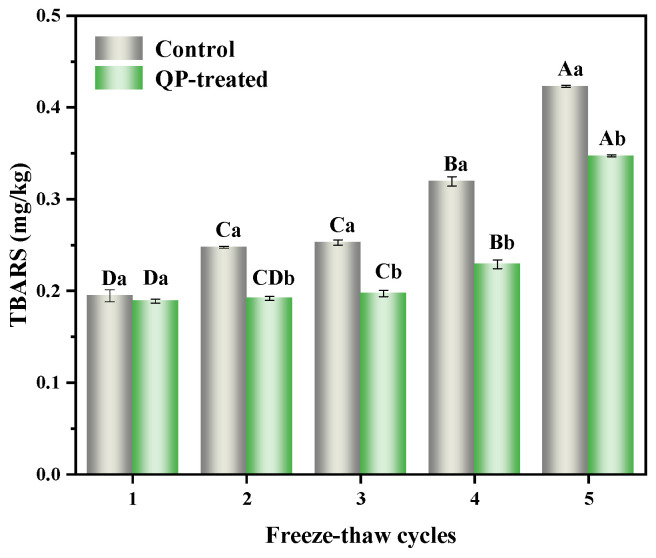
Effect of QP addition on TBARS of low-sodium pork patties during different F-T cycles. Note: QP refers to quinoa protein. Different lowercase letters indicate a significant difference between the treatment groups within the same F-T cycle (*p* < 0.05); different uppercase letters indicate a significant difference between the F-T cycles within the same treatment group (*p* < 0.05).

**Figure 4 foods-13-00522-f004:**
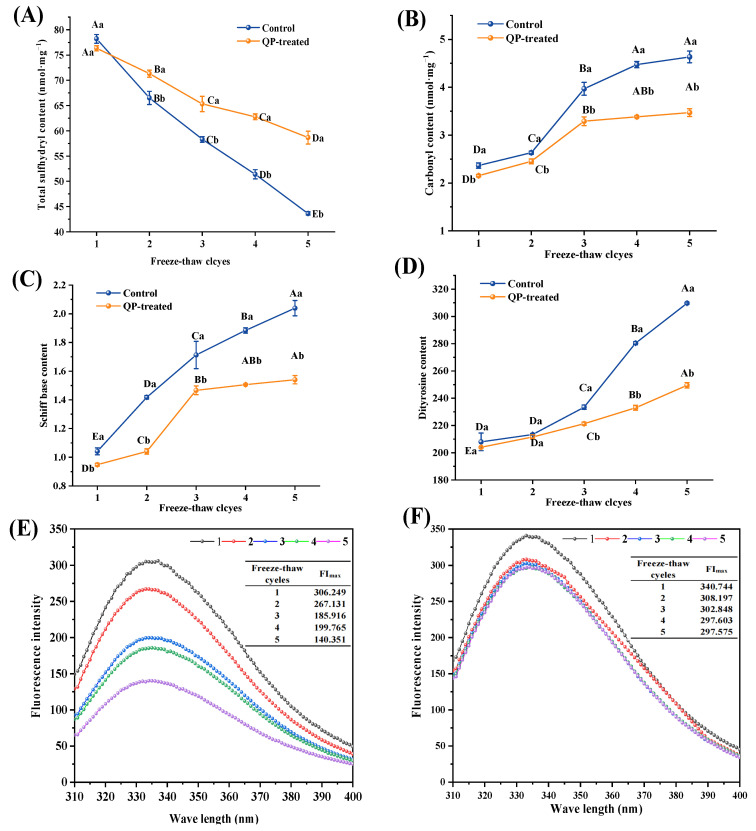
Effect of QP addition on the total sulfhydryl content (**A**), carbonyl content (**B**), Schiff base content (**C**), dityrosine content (**D**), fluorescence intensity of the control group (**E**), and fluorescence intensity of the QP-treated (**F**) group of myofibrillar protein from low-sodium pork patties during different F-T cycles. Note: QP refers to quinoa protein. Different lowercase letters indicate a significant difference between the treatment groups within the same F-T cycle (*p* < 0.05); different uppercase letters indicate a significant difference between the F-T cycles within the same treatment group (*p* < 0.05).

**Figure 5 foods-13-00522-f005:**
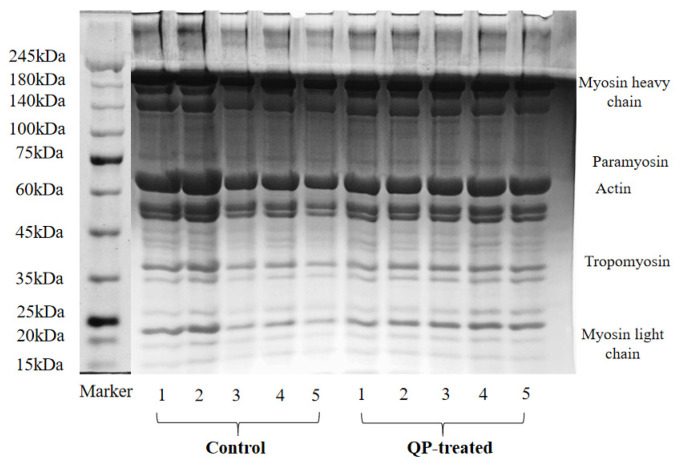
Changes in SDS-PAGE pattern of myofibrillar protein from low-sodium pork patties during different F-T cycles. Note: QP refers to quinoa protein.

**Figure 6 foods-13-00522-f006:**
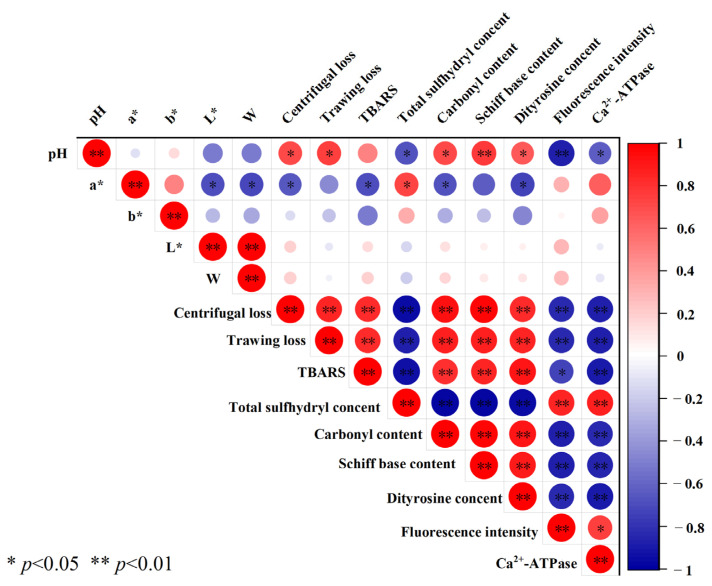
The correlation analysis of storage quality indexes and protein physicochemical indexes of low-sodium pork patties during five F-T cycles. “*” indicates that the difference between different indicators is significant (*p* < 0.05), and “**” indicates that the difference between different indicators is extremely significant (*p* < 0.01).

**Table 1 foods-13-00522-t001:** The changes in the color of pork patties during different F-T cycles.

F-T Cycles	*L**-Value		*a**-Value		*b**-Value		*W**-Value	
	Control	QP-Treated	Control	QP-Treated	Control	QP-Treated	Control	QP-Treated
1	50.15 ± 0.72 ^Da^	44.42 ± 0.35 ^Cb^	5.61 ± 0.33 ^Aa^	6.06 ± 0.09 ^Aa^	9.72 ± 0.47 ^Db^	12.52 ± 0.34 ^Aa^	48.90 ± 0.70 ^Da^	42.71 ± 0.42 ^Bb^
2	54.32 ± 0.51 ^Ca^	45.43 ± 0.30 ^Bb^	4.83 ± 0.17 ^Bb^	5.72 ± 0.08 ^ABa^	13.40 ± 0.05 ^Aa^	13.04 ± 0.65 ^Aa^	52.15 ± 0.48 ^Ca^	43.60 ± 0.19 ^Bb^
3	56.60 ± 0.19 ^Ba^	49.50 ± 0.34 ^Ab^	4.24 ± 0.22 ^Cb^	5.56 ± 0.28 ^Ba^	11.31 ± 0.43 ^Cb^	13.00 ± 0.05 ^Aa^	54.95 ± 0.15 ^Ba^	47.56 ± 0.36 ^Ab^
4	57.45 ± 0.38 ^Aa^	49.79 ± 0.10 ^Ab^	4.16 ± 0.11 ^Ca^	4.18 ± 0.07 ^Ca^	12.32 ± 0.15 ^Ba^	10.58 ± 0.22 ^Bb^	55.51 ± 0.35 ^Ba^	48.52 ± 0.14 ^Ab^
5	59.01 ± 0.90 ^Aa^	49.42 ± 0.18 ^Ab^	3.95 ± 0.09 ^Ca^	3.73 ± 0.13 ^Da^	9.73 ± 0.22 ^Da^	10.32 ± 0.49 ^Ba^	57.68 ± 0.92 ^Aa^	48.24 ± 0.28 ^Ab^

Note: QP refers to quinoa protein. Different lowercase letters indicate a significant difference between the treatment groups within the same F-T cycle (*p* < 0.05); different uppercase letters indicate a significant difference between the F-T cycles within the same treatment group (*p* < 0.05).

## Data Availability

The data presented in this study are available in the article.

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
