# Peer review of "The Protection of Quinoa Protein on the Quality of Pork Patties during Freeze–Thaw Cycles: Physicochemical Properties, Sensory Quality and Protein Oxidative"

_foods, 2024, doi:10.3390/foods13040522_

Round 1
Reviewer 1 Report
Comments and Suggestions for Authors
This study determines the potential of using quinoa protein as a protectant for pork patties affected by multiple freeze-thaw cycles by monitoring their physicochemical properties, protein oxidation, and sensory evaluation, compared to the control. The scope is interesting. The MS exhibits clear logic and a certain level of novelty, and the relevant results carry guiding significance for improving the quality of meat processing. However, the experimental design, particularly the statistical analysis, raises queries, and the relevant results are not adequately expressed in the MS. Additionally, the relevant analysis lacks depth. Therefore, the MS needs major revisions. After significant revisions, I believe it can be published in this journal. Specific questions and comments are as follows:
General comments:
- There is a difference in the terms used to describe the overall parameters determined in this study between the topic and abstract. The topic refers to 'protein oxidation,' while the abstract uses 'MP properties.' I recommended using consistent terminology for better clarity. If the study involves measurements beyond just oxidation and includes other properties, I suggested using the term 'properties' to encompass the broader scope, as 'MP properties' is wider than 'oxidation'.
ABSTRACT
- I suggest to improve the abstract. The empirical data or interesting outcomes (in details likes expressed in number) should be added, in couple with the statistical analysis.
INTRODUCTION
- How’s protein can act as a protectant? This should be described either in the introduction or discussion. This can be help linked to the structure/composition of quinoa act as protectant. Please increase the coherence of this context.
- I suggested that the author include examples of protein protectants from previous research and highlight their effectiveness. This addition would strengthen the introduction that they can be used as alternative protectants, particularly without adversely affecting product quality.
- Line72-76: What does the author mean, in the previous studies? It seem like this current study?? Please clarify this point. How’s different among them. Author should re-write this context (paragraph), stated the gap from previous finding, which brings to this current study.
MATERIAL AND METHODS
- How much of the samples and how’s variation of sample used in the study. Did the samples obtained from the same market and brought only once? Normally, we recognize the importance of sampling to ensure a representative sample of the population. This is, therefore, allocated the sample collecting, for at least 2-3 lots.
- For this study, the actual temperatures of samples during freezing or thawing were crucial. It seems like the author monitored them; however, these details were not explicitly stated in the results and discussion section. Fluctuations in temperature could potentially lead to misleading conclusions. Thus, I suggest providing this information, perhaps in the supplementary data.
- For sensory evaluation, why ten postgraduate students were selected? How’s about the training process? The expertise of panelists are highly relevant to sensory quality evaluation of chicken breast.
- For the sample preparation before conducting sensory evaluation, please provide details on how the samples were cooked at 200°C for 20 minutes. Additionally, I recommend placing the sensory evaluation either as the first or last parameter, as it differs significantly from other chemical and physical analyses.
- For the statistical analysis, I am curious about why the author chose to use one-way ANOVA. Given the presence of two factors (control/QPtreated and the F-T cycles), could you elaborate on the rationale behind this choice? Furthermore, when comparing control and QP-treated, considering that there were only two samples, it seems that ANOVA may not be suitable. Could you please clarify or correct this point?
RESULTS AND DISCUSSION
- For the pH, the pH of the control continuously increased with an increasing number of F-T cycles, while a decrease in pH was observed after the 5th F-T cycle in the QP-treated samples. Could you please discuss the reasons or possibilities behind this observation? Additionally, how does this relate to the quality of the patties or any other parameters?
- For Figure 1, I suggest separating it into two graphs (1a and 1b) for easier understanding.
- For color, could you provide more information about what the whiteness (w*) parameter refers to? Please discuss the difference obtained among all samples (both between control and added quinoa-samples, as well as when increase the F-T cycle)
- Was there a statistical analysis conducted for the sensory evaluation? The results were not presented, and the author did not mention it in the discussion. This is crucial, considering it is an important criterion used to determine consumer acceptance of the developed product. Furthermore, for the sensory scores, what specific score or level is used as the criterion to determine whether the product is considered acceptable or not? Please provide a reference or include information related to this criterion in the results and discussion.
- Line461-462: The bands of the control group weakened in as F-T cycles increased??? Is can referred to the oxidative degradation???? Why???
- Line 478-479: To further explore the effect of QP addition on the storage quality and antioxidant
properties of pork patties during different F-T cycles >>> Is this the further study? I recommend rephrasing the sentence to emphasize that additional insights were gained through relationship analysis.
- Overall, the authors present a large number of structural characterization results but do not provide a detailed analysis process and related conclusions. The conclusions related to the changes mentioned in the manuscript are not typical. The authors should conduct a more in-depth analysis and exploration of this aspect. An extended discussion of the effect of both variables may add novelty to the work.
Author Response
Responses to the Comments of Reviewer
Dear Reviewer,
Thank you for your letter and comments on our manuscript entitled “The protection of quinoa protein on the quality of pork patties during freeze-thaw cycles: physicochemical properties, sensory quality and protein oxidation”. Those comments are all valuable and very helpful for revising and improving our paper, as well as the important guiding significance to our researches. We have studied comments carefully and have made correction which we hope meet with approval. Revised portion are marked in red in the paper. The main corrections in the paper and the responds to the reviewer’s comments are as following:
Reviewer: 1
Comments:
This study determines the potential of using quinoa protein as a protectant for pork patties affected by multiple freeze-thaw cycles by monitoring their physicochemical properties, protein oxidation, and sensory evaluation, compared to the control. The scope is interesting. The MS exhibits clear logic and a certain level of novelty, and the relevant results carry guiding significance for improving the quality of meat processing. However, the experimental design, particularly the statistical analysis, raises queries, and the relevant results are not adequately expressed in the MS. Additionally, the relevant analysis lacks depth. Therefore, the MS needs major revisions. After significant revisions, I believe it can be published in this journal. Specific questions and comments are as follows
Response: Thank you sincerely for acknowledging our work. We highly appreciate your valuable feedback. The manuscript has been thoroughly revised, and we have incorporated the necessary changes as per your comments. Below, you will find our responses along with the corresponding revisions made in the updated version.
General comments:
- There is a difference in the terms used to describe the overall parameters determined in this study between the topic and abstract. The topic refers to 'protein oxidation,' while the abstract uses 'MP properties.' I recommended using consistent terminology for better clarity. If the study involves measurements beyond just oxidation and includes other properties, I suggested using the term 'properties' to encompass the broader scope, as 'MP properties' is wider than 'oxidation'.
Response: Thank you for your valuable comments. We highly appreciate your suggestion, as it aligns well with the focus of this study which primarily investigates the oxidation property of MP. Therefore, we have opted to replace the term "MP properties" with "MP oxidation", making necessary modifications highlighted in red within this paper. (Line 15 and 35 in the reworked manuscripts)
ABSTRACT
- I suggest to improve the abstract. The empirical data or interesting outcomes (in details likes expressed in number) should be added, in couple with the statistical analysis.
Response: Thank you very much for your valuable comments, we have rewritten the abstract and changed the red marking in the "Abstract".
The present study investigated the impact of quinoa protein (QP) on the physicochemical properties, sensory quality, and oxidative stability of myofibrillar protein (MP) in pork patties during five freeze-thaw (F-T) cycles. It was observed that repeated F-T cycles resulted in a deterioration of pork patty quality; however, the incorporation of QP effectively mitigated these changes. Throughout the F-T cycles, the sensory quality of the QP-treated group consistently surpassed that of the control group. After five F-T cycles, the thiobarbituric acid reactive substances (TBARS) content in the control group was measured at 0.423 mg/kg, whereas it significantly decreased to 0.347 mg/kg in the QP-treated group (p < 0.05). Furthermore, QP inclusion led to a decrease in pH and an increase in water-holding capacity (WHC) within pork patties. Following five F-T cycles, Ca2+-ATPase activity exhibited a significant increase by 11.10% in the QP-treated group compared to controls (p < 0.05). Additionally, supplementation with QP resulted in elevated total sulfhydryl content and reduced carbonyl content, schiff base content, and dityrosine content within myofibrillar proteins (MPs), indicating its inhibitory effect on MP oxidation. In particular, after five F-T cycles, total sulfhydryl content reached 58.66 nmol/mL for the QP-treated group significantly higher than that observed for controls at 43.65 nmol/mL (p < 0.05). While carbonyl content increased from 2.37 nmol/mL to 4.63 nmol/mL between first and fifth F-T cycle for controls; it only rose from 2 .15 nmol/mL to 3.47 nmol/mL in the QP-treated group.The endogenous fluorescence levels were significantly higher (p < 0.05) in the QP-treated group compared to controls. In conclusion, the addition of QP enhanced the quality of pork patties and effectively inhibited the oxidative denaturation of MP during F-T cycles.
INTRODUCTION
- How’s protein can act as a protectant? This should be described either in the introduction or discussion.This can be help linked to the structure/composition of quinoa act as protectant. Please increase the coherence of this context.
Response: We greatly appreciate your valuable comments. The utilization of proteins as cryoprotectants is based on two fundamental principles: firstly, these proteins and their derivatives possess antioxidant properties; secondly, they have the ability to bind to existing ice crystals, thereby inhibiting their further growth and reducing mechanical damage caused by large ice crystals in food. Consequently, their incorporation effectively prevents the oxidative cross-linking aggregation denaturation of MP in food. (Cao, H., Zheng, X., Liu, H., Yuan, M., Ye, T., Wu, X., … Xu, F. (2020). Cryo-protective effect of ice-binding peptides derived from collagen hydrolysates on the frozen dough and its ice-binding mechanisms. Lwt, 131, Article 109678. https://doi.org/ 10.1016/j.lwt.2020.109678.) (Line 73~79 in the reworked manuscripts)
- I suggested that the author include examples of protein protectants from previous research and highlight their effectiveness. This addition would strengthen the introduction that they can be used as alternative protectants, particularly without adversely affecting product quality.
Response: Thank you for your valuable comments. We had added examples of previous studies on protein cryoprotectants to the manuscript and found that the addition of protein cryoprotectants had no adverse effect on product quality. Relevant studies are listed below and have been added to the manuscript in red. ((Line 79~85 in the reworked manuscripts)
Wang et al. found that chickpea inhibited the denaturation and oxidation of MP during freeze-thaw cycles, resulting in increased MP solubility and enhanced Ca2+-ATPase activity. In addition, gels prepared from chickpea protein mixed with surimi also exhibited a more stable microstructure.(Wang C, Rao J, Li X, et al. Chickpea protein hydrolysate as a novel plant-based cryoprotectant in frozen surimi: Insights into protein structure integrity and gelling behaviors[J]. Food Research International, 2023, 169: 112871.)Lin et al. found that gill hydrolysate had antioxidant cryoprotective effects on surimi, comparable to those of commercial cryoprotectants, and could be used as an alternative to sugar cryoprotectants. (Lin J, Hong H, Zhang L, et al. Antioxidant and cryoprotective effects of hydrolysate from gill protein of bighead carp (Hypophthalmichthys nobilis) in preventing denaturation of frozen surimi[J]. Food Chemistry, 2019, 298: 124868.)
- Line72-76: What does the author mean, in the previous studies? It seem like this current study?? Please clarify this point. How’s different among them. Author should re-write this context (paragraph), stated the gap from previous finding, which brings to this current study.
Response: Thank you again for your positive comments and valuable suggestions to improve the quality of our manuscript. In the study preceding this experiment, we explored the effect of different additions of quinoa protein on the quality of pork patties and their digestive characteristics. It was found that quinoa protein added at 6% had the most significant effect on improving the quality and digestive characteristics of pork patties. In this study, pork patties made with 6% quinoa protein addition were used as a experimental group, and pork patties without quinoa protein addition were used as a control group to investigate the effect of quinoa protein on the cryoprotective effect of repeated freeze-thawing of pork patties and their MP. In the manuscript, we have modified this part of the text, and the modifications are marked in red. (Line 94~103 in the reworked manuscripts)
MATERIAL AND METHODS
- How much of the samples and how’s variation of sample used in the study. Did the samples obtained from the same market and brought only once? Normally, we recognize the importance of sampling to ensure a representative sample of the population. This is, therefore, allocated the sample collecting, for at least 2-3 lots.
Response: Thank you for your valuable comments. Our samples were taken from the same company and the same batch of Jinfen White Pigs, but we conducted this experiment in batches and the indicators measured later were repeated more than 3 times, which could guarantee the reliability of the sampling.
- For this study, the actual temperatures of samples during freezing or thawing were crucial. It seems like the author monitored them; however, these details were not explicitly stated in the results and discussion section. Fluctuations in temperature could potentially lead to misleading conclusions. Thus, I suggest providing this information, perhaps in the supplementary data.
Response: Thank you for your valuable comments. In this study, the freezing temperature of the pork patties was maintained at 18℃, and the center temperature of the pork patties was considered fully thawed when it reached 4℃, at which point thawing was halted. To prevent any interference from temperature variations in the study, we have explicitly specified in "2. Materials and Methods" the temperatures at which all experiments should be conducted and highlighted them in red.
- For sensory evaluation, why ten postgraduate students were selected? How’s about the training process? The expertise of panelists are highly relevant to sensory quality evaluation of chicken breast.
Response: We are sorry that we didn't make this clear. Our sensory evaluation test was conducted at the Sensory Analysis Laboratory of Food Science and Engineering, Shanxi Agricultural University. Most of the postgraduate students here have been trained for a long time in specialized courses related to sensory evaluation. Before the start of the trial, we all provided each volunteer with a one-week training on the subject. Our sensory evaluation team members had sufficient expertise to conduct the sensory evaluation for this trial.
- For the sample preparation before conducting sensory evaluation, please provide details on how the samples were cooked at 200°C for 20 minutes. Additionally, I recommend placing the sensory evaluation either as the first or last parameter, as it differs significantly from other chemical and physical analyses.
Response: Thank you for your valuable comments. Pork patties are heated as follows, and we have added the details of heating to "2.8 Sensory Assessment".
Heat the pork patties in an oven at a temperature of 200°C for 20 min, keeping a close eye on them during the heating process to make sure they are not overcooked or burnt. At the end of the heating process, allow the pork patties to cool slightly and then cut them evenly into 2 x 2 cm pieces. (Line 181~184 in the reworked manuscripts)
Furthermore, we also have included sensory assessment as the first parameter, changes are highlighted in red. (Line 252~274 in the reworked manuscripts)
- For the statistical analysis, I am curious about why the author chose to use one-way ANOVA. Given the presence of two factors (control/QPtreated and the F-T cycles), could you elaborate on the rationale behind this choice? Furthermore, when comparing control and QP-treated, considering that there were only two samples, it seems that ANOVA may not be suitable. Could you please clarify or correct this point?
Response: We sincerely apologize for our oversight. In the statistical analysis, we employed a two-way ANOVA approach. The figure and table note in the manuscript have been revised to state: "Different lowercase letters indicate a significant difference between treatment groups within the same F-T cycle (p < 0.05); different uppercase letters indicate a significant difference between F-T cycles within the same treatment group (p < 0.05)." We have made the necessary adjustments by changing from one-way ANOVA to two-way ANOVA in our resubmitted manuscript, and we appreciate your assistance with this correction.
Analysis of variance (ANOVA) is a statistical method utilized for examining the impact of various factors on the dependent variable in data analysis. Its fundamental principle involves decomposing total variation into within-group and between-group variations, subsequently determining whether there exists a significant effect of independent variables on dependent variables through comparing these variations' magnitudes. ANOVA serves as a widely adopted statistical tool that aids in assessing independent variable effects, comparing component differences, controlling for errors, enhancing statistical efficacy, and providing guidance for further research.
RESULTS AND DISCUSSION
- For the pH, the pH of the control continuously increased with an increasing number of F-T cycles, while a decrease in pH was observed after the 5th F-T cycle in the QP-treated samples. Could you please discuss the reasons or possibilities behind this observation? Additionally, how does this relate to the quality of the patties or any other parameters?
Response: Thank you for your valuable comments. The pH of the QP-treated group decreased in the 5th F-T cycle, which can be attributed to the continuous degradation of proteins in the meat caused by repeated F-T cycles, resulting in a continuous production of various amino acids. Consequently, there was a higher content of acidic amino acids compared to basic amino acids, leading to a lower pH. The revised manuscript revisits the reasons behind this outcome and highlights the changes made. The decrease in pH at this stage indicates an altered state of protein breakdown and oxidation of the pork patties due to QP treatment, contributing to their aging and deterioration in quality (Line 283~287 in the reworked manuscripts).
- For Figure 1, I suggest separating it into two graphs (1a and 1b) for easier understanding.
Response: Thank you for your valuable comments. We have divided the water holding capacity into two graphs as per your comments, Figure 2C and Figure 2D.(Line 291 in the reworked manuscripts)
- For color, could you provide more information about what the whiteness (w*) parameter refers to? Please discuss the difference obtained among all samples (both between control and added quinoa-samples, as well as when increase the F-T cycle)
Response: Thank you for your valuable comments. We have discussed the W* values for the QP-treated and control groups in different F-T cycles, and this discussion is highlighted in red sections of the 'Color in Results and Discussion' within the manuscript (lines 337-340 in the revised manuscripts)."
- Was there a statistical analysis conducted for the sensory evaluation? The results were not presented, and the author did not mention it in the discussion. This is crucial, considering it is an important criterion used to determine consumer acceptance of the developed product. Furthermore, for the sensory scores, what specific score or level is used as the criterion to determine whether the product is considered acceptable or not? Please provide a reference or include information related to this criterion in the results and discussion.
Response: We apologize for the lack of clarity in our previous statement. We would like to clarify that the sensory evaluation of pork patties did undergo statistical analysis. In the revised manuscript, we have made revisions to the wording of the sensory assessment, which are highlighted in red. The sensory scores were obtained using a single nine-point scale as described by Kumari et al. (2023). This scale ranges from 1 (strong dislike) to 9 (strong liking), with a score above 5 indicating consumer acceptability. To address expert opinions, we have included this reference standard in both the "Sensory evaluation" section of Results and Discussion.
(Kumari, S., Alam, A. N., Hossain, M. J., Lee, E. Y., Hwang, Y. H., & Joo, S. T. (2023). Sensory Evaluation of Plant-Based Meat: Bridging the Gap with Animal Meat, Challenges and Future Prospects. Foods, 13(1), 108.)
- Line461-462: The bands of the control group weakened in as F-T cycles increased??? Is can referred to the oxidative degradation???? Why???
Response: We apologize for not making this point clear. changes in SDS-PAGE protein bands by themselves are usually not directly indicative of the level of protein oxidation. sds-PAGE is a technique for separating proteins based on their molecular weights, and is primarily used for protein isolation, characterization, and quantitation. However, in some cases, SDS-PAGE can provide indirect information about protein oxidation. For example, if a protein undergoes oxidation, this may result in a change in its molecular weight or the formation of aggregates, which may show different bands or changes in the intensity of the bands on the SDS-PAGE chromatogram. As also mentioned in the following articles, a weakening of protein bands indicates degradation of the protein, which in turn indicates increased protein oxidation.
Zhang, D., Li, H., Emara, A. M., Wang, Z., Chen, X., & He, Z. (2020). Study on the mechanism of KCl replacement of NaCl on the water retention of salted pork. Food Chemistry, 332, 127414.
- Line 478-479: To further explore the effect of QP addition on the storage quality and antioxidant properties of pork patties during different F-T cycles >>> Is this the further study? I recommend rephrasing the sentence to emphasize that additional insights were gained through relationship analysis
Response: Thank you for your valuable comments. We have discussed this sentence and it really doesn't fit and we have revised the sentence. Revised sentence to read "In order to investigate the relationship between the quality characteristics of pork patties and the degree of MP oxidation, a correlation analysis was conducted between the physicochemical characteristics of pork patties and the MP oxidation indexes." This sentence has been highlighted in red in the section "3.13. Analysis of relationship".(Line 514~516 in the reworked manuscripts)
- Overall, the authors present a large number of structural characterization results but do not provide a detailed analysis process and related conclusions. The conclusions related to the changes mentioned in the manuscript are not typical. The authors should conduct a more in-depth analysis and exploration of this aspect. An extended discussion of the effect of both variables may add novelty to the work.
Response: Thank you for your valuable comments. In response to your comments, we have reworded the "Conclusion" section.
Reviewer 2 Report
Comments and Suggestions for Authors
Congratulations for this manuscript. The study of the impact of quinoa protein (QP) on physicochemical properties, sensory quality and myofibrillar protein (MP) properties of pork patties during five freeze-thaw16 (F-T) cycles it may be of great interest for the food industry. The results will have a potential transfer and application in food processing.
Introduction, material and methods and results have been well described. Nevertheless, I would recommmed the authors to change the results section name fpor "results and discussion".
Before it is finally accepted I would suggest the authors to clarify certain aspects like:
-How is that a manuscript can be supported by 3 different foundings. Did all these 3 foundings support the same objective
QP had a positive cryoprotective effect on pork patties during the freeze-thaw cycle.
Why do authors state in the conclusions that "This natural cryoprotectant was an excellent sustainable alternative source of animal protein"?. Why do you consider QP sustainable?. Is this term related or a result of your paper?. On what basis do you state this?
The following conclusion "This discovery had significant implications for the development and preservation of meat products" could be better develop in relation with the food industry more than to the meat products. The knowledge transfer and potential application should be better presneted.
The following conclusion "It also provided theoretical support for the possibility of substituting animal protein with plant protein" may not be based on the results of this study. If so, authors should describe this better.
Author Response
Responses to the Comments of Reviewer
Dear Reviewer,
Thank you for your letter and comments on our manuscript entitled “The protection of quinoa protein on the quality of pork patties during freeze-thaw cycles: physicochemical properties, sensory quality and protein oxidation”. Those comments are all valuable and very helpful for revising and improving our paper, as well as the important guiding significance to our researches. We have studied comments carefully and have made correction which we hope meet with approval. Revised portion are marked in red in the paper. The main corrections in the paper and the responds to the reviewer’s comments are as following:
Reviewer: 2
The study of the impact of quinoa protein (QP) on physicochemical properties, sensory quality and myofibrillar protein (MP) properties of pork patties during five freeze-thaw16 (F-T) cycles it may be of great interest for the food industry. The results will have a potential transfer and application in food processing.
Response: Thank you very much for your recognition of our work. We really appreciate your valuable comments.
- Introduction, material and methods and results have been well described. Nevertheless, I would recommmed the authors to change the results section name fpor "results and discussion".
Response: Thank you for your valuable comments. We have changed the word "results" to "results and discussion", with the changes highlighted in red in the manuscript. (Line 251 in the reworked manuscripts)
- Before it is finally accepted I would suggest the authors to clarify certain aspects like:
Response: Thank you for your valuable comments.Thank you very much for your input. We have revised and replied to the manuscript one by one according to your comments, and the specific replies are as follows.
- How is that a manuscript can be supported by 3 different foundings. Did all these 3 foundings support the same objective
Response: Thank you for your valuable feedback. The completion of this manuscript was supported by all three funds involved in this trial. The pork utilized in this study originated from Jinfen white pigs, which is the main focus of the Shanxi Provincial Agricultural Major Technology Synergistic Extension Project (2023XTTG01) that aims to improve breeding and meat processing techniques for Jinfen white pigs. The research conducted in this experiment aligns with the objectives of this project. The utilization of quinoa protein to produce pork patties in this study is primarily due to its high nutritional value and functional properties, which are integral components of the research content covered by the Functional Food Technology System Construction Project of Shanxi Province (2023CYJSTX10-03). Additionally, financial support for reagent determination and related indicators used in this experiment was provided by the Natural Science Research Projects of Shanxi Province (20210302123400).
- QP had a positive cryoprotective effect on pork patties during the freeze-thaw cycle.
Response: Thank you for your valuable comments. We have reworded this sentence, and the changes are highlighted in red in lines 551-553 of the reworked manuscript.
- Why do authors state in the conclusions that "This natural cryoprotectant was an excellent sustainable alternative source of animal protein"? Why do you consider QP sustainable? Is this term related or a result of your paper? On what basis do you state this?
Response: We are so sorry for the unclear expression. We found QP to be an excellent alternative source of sustainable animal protein by referring to the relevant literature, which is referenced below.
- Dakhili, S.; Abdolalizadeh, L.; Hosseini, S.M.; Shojaee-Aliabadi, S.; Mirmoghtadaie, L. Quinoa Protein: Composition, Structure and Functional Properties. Food Chemistry 2019, 299, 125161, doi:10.1016/j.foodchem.2019.125161
- Wang S, Miao S, Sun D W. Modifying structural and techno-functional properties of quinoa proteins through extraction techniques and modification methods[J]. Trends in Food Science & Technology, 2023: 104285.
- The following conclusion "This discovery had significant implications for the development and preservation of meat products" could be better develop in relation with the food industry more than to the meat products. The knowledge transfer and potential application should be better presneted.
Response: Thank you for your valuable feedback. We have replaced the term "meat products" with "the food industry" throughout the manuscript. The revised sections are highlighted in red, specifically lines 71 to 74 and lines 551 to 552 in the reworked manuscripts.
- The following conclusion "It also provided theoretical support for the possibility of substituting animal protein with plant protein" may not be based on the results of this study. If so, authors should describe this better.
Response: Thank you for your valuable comments. This sentence does have some problems as you mentioned, and we have reworked it, with the changes highlighted in red in the conclusion section.(Line 558~560 in the reworked manuscripts)
Round 2
Reviewer 1 Report
Comments and Suggestions for Authors
The authors conducted the required revisions as suggested before.